# Interactive Submodular Bandit

**Lin Chen**[1,2], **Andreas Krause**[3], **Amin Karbasi**[1,2]
[1] Department of Electrical Engineering, [2] Yale Institute for Network Science, Yale University
[3] Department of Computer Science, ETH Zürich
{lin.chen, amin.karbasi}@yale.edu, krausea@ethz.ch

## Abstract

In many machine learning applications, submodular functions have been used as a model for evaluating the utility or payoff of a set such as news items to recommend, sensors to deploy in a terrain, nodes to influence in a social network, to name a few. At the heart of all these applications is the assumption that the underlying utility/payoff function is *known* a priori, hence maximizing it is in principle possible. In many real life situations, however, the utility function is not fully known in advance and can only be estimated via interactions. For instance, whether a user likes a movie or not can be reliably evaluated only after it was shown to her. Or, the range of influence of a user in a social network can be estimated only after she is selected to advertise the product. We model such problems as an *interactive* submodular bandit optimization, where in each round we receive a context (e.g., previously selected movies) and have to choose an action (e.g., propose a new movie). We then receive a noisy feedback about the utility of the action (e.g., ratings) which we model as a submodular function over the context-action space. We develop SM-UCB that efficiently trades off exploration (collecting more data) and exploration (proposing a good action given gathered data) and achieves a $O(\sqrt{T})$ regret bound after $T$ rounds of interaction. More specifically, given a bounded-RKHS norm kernel over the context-action-payoff space that governs the smoothness of the utility function, SM-UCB keeps an upper-confidence bound on the payoff function that allows it to asymptotically achieve no-regret. Finally, we evaluate our results on four concrete applications, including movie recommendation (on the MovieLense data set), news recommendation (on Yahoo! Webscope dataset), interactive influence maximization (on a subset of the Facebook network), and personalized data summarization (on Reuters Corpus). In all these applications, we observe that SM-UCB consistently outperforms the prior art.

## 1 Introduction

Interactive learning is a modern machine learning paradigm that has recently received significant interest in both theory and practice [15, 14, 7, 6]. In this setting, the learning algorithm engages in a two-way dialog with the environment (e.g., users) by performing actions and receiving a response (e.g., like or dislike) for each action. Interactive learning has led to substantial performance improvement in a variety of machine learning applications [43, 13], including clustering [4, 25, 1], classification [46, 10], language learning [48], decision making [26], and recommender systems [28], to name a few.

At a high level, interactive learning can be cast as a dynamic optimization problem with a *known* utility/payoff function where the goal is to achieve an objective whose value depends on the selected actions, their responses, and the state of the environment. In many practical settings, the utility functions are submodular, stating (informally) that the payoff of performing an action earlier is more

than performing it later. In fact, rigorous treatment of submodularity in interactive settings led to strong theoretical guarantees on the performance of greedy policies [19, 22].

In this paper, we go one step further and consider scenarios where the exact form of the submodular payoff function is not completely known and hence needs to be estimated through interactions. This problem is closely related to the contextual multi-armed bandit [38, 9, 36] where for a sequence of $T$ rounds, we receive a payoff function along with some side information or context (e.g., user's features), based on which we have to choose an action (e.g., proposing an item) and then a noisy feedback about the obtained payoff is revealed (e.g., rating of the proposed item). The goal is to minimize the regret of not selecting the optimal action due to the uncertainty associated with the utility function. The interactive contextual bandit generalizes this setting by allowing to interact with a payoff function multiple times, where each time we need to take a new action based on both the context and previously taken actions. The regret is then defined in terms of the difference between the utility of the best set of actions that we could have chosen versus the ones that are actually selected. In this paper, we further assume that the marginal payoffs of actions show diminishing returns. This problem, which we call interactive submodular bandit, appears in many practical scenarios, including:

- **Interactive recommender system.** The goal is to design a recommender system that interacts with the users in order to elicit and satisfy their preferences. In our approach, we model the utility of a set of items as an *unknown* submodular objective function that the recommender systems aims to maximize. In each round of interaction, the recommender system decides which item should be presented to the user, given the previously proposed items to this or similar users (affinity between users, if this side information exists, can be leveraged to enhance recommendation performance). Since the users' preferences are unknown, the recommender system can only gather information about the users through the feedback they provide in terms of ratings. A successful recommender system should be able to minimize the total regret accumulated over $T$ iterations with users.

- **Interactive influence maximization.** Influence spread maximization addresses the problem of selecting the most influential source nodes of a given size in a diffusion network [51]. A diffusion process that starts with those source nodes can potentially reach the greatest number of nodes in the network. Under many diffusion models, the expected total number of influenced people is a submodular function of the seed subjects [29, 20]. In a natural interactive variant of this problem, users may be recruited in a sequential manner [42, 5] where a new user is selected once we fully observe the extent to which the current seed users influenced people. Note that finding the optimal set of source nodes in a diffusion network depends dramatically on the underlying dynamics of the diffusion. Again, very often in practice, we are faced with the dilemma of estimating the underlying diffusion parameters through interactively selecting the nodes while at the same time trying to maximize the influence.

- **Interactive coverage.** Coverage problems arise naturally in many applications [32, 33]. Consider an environmental monitoring task, for instance, where sensors are placed in the Alps to better predict floods, landslides and avalanches [8]. Similarly, Wi-Fi hotspots are carefully arranged to cover every corner of a floor. However, it is likely that the actual coverage of a device is uncertain before deployment due to unknown conditions present in the environment. Hence, we might need to install devices in a sequential manner after we observe the actual coverage of the ones already installed [19, 22].

Without any assumptions about the smoothness of the payoff function, no algorithm may be able to achieve low regret [31]. Thus, in our setting, we make a crucial yet very natural assumption that the space of context-action-payoff has low complexity, quantified in terms of the Reproducing Kernel Hilbert Space (RKHS) norm associated with some kernel [24]. We then show that SM-UCB, an upper-confidence-bound based algorithm achieves an $O(\sqrt{T})$ regret. We also evaluate the performance of SM-UCB on four real-world applications, including movie recommendation [40], news recommendation [16], interactive influence maximization [42, 34], and personalized data summarization [39].

## 2 Problem Formulation

As we stated earlier, many utility or payoff functions we encounter in machine learning applications are submodular. As a reminder, a set function $f$ is called *submodular* if for all $A \subseteq B \subseteq \Omega$ we have

$$f(A) + f(B) \geq f(A \cap B) + f(A \cup B).$$

An equivalent definition of submodularity that shows better the *diminishing returns* property is as follows: for all $A \subseteq B \subseteq \Omega$ and any element $e \notin B$ we have

$$f(A \cup \{e\}) - f(A) \geq f(B \cup \{e\}) - f(B).$$

We also denote the *marginal gain* of an element $e$ to a set $A$ by $\Delta(e|A) \triangleq f(A \cup \{e\}) - f(A)$. The function $f$ is called *monotone* if for all $A \subseteq B$ we have $f(A) \leq f(B)$.

In this paper, we consider a sequential decision making process for a horizon of $T$ time steps, where at each round $i$, a monotone submodular function $f_i$, is (partially) revealed. Let us first consider the simple bandit problem [50] where we need to select an item (or an arm) $e$ from the set of items $\Omega$ such that $f_i(e)$ is maximized. After the item $e$ is selected, the payoff $f_i(e)$ is revealed. Since $f_t$'s are not known in advance, the goal is to minimize the accumulated regret over $T$ rounds, for not choosing the optimum items. Contextual submodular bandit generalizes the aforementioned setting by allowing to receive side information $\phi_i$ (also called context) in each round $i$ [31]. But still the goal is to select a *single* item $e$ such that $f_{\phi_i}(e)$ is maximized. In the *interactive* contextual submodular bandit, the focus of this paper, we may encounter the same valuation function $f_\phi$, with its associate context $\phi$, multiple times over the time horizon $T$. Here instead, at each round, we need to propose a *new* item that maximizes the *marginal* payoff given the ones we selected in the previous encounters. Therefore, we are sequentially building up subsets of items that maximize the payoff for each separate function $f_\phi$. For instance, a recommender system may interact with a user (or a number of users) multiple times. In each interaction, it has to recommend a new item while taking into account what it has recommended in previous interactions.

More formally, let us assume that we encounter $m \leq T$ distinct functions $f_\phi$, in an arbitrary order, over the time horizon $T$, i.e., $\phi \in \{\phi_1, \ldots, \phi_m\}$. We denote the arriving ordered sequence by $f_1, f_2, \ldots, f_T$ where for each round $i$, we have $f_i \in \{f_{\phi_1}, \ldots, f_{\phi_m}\}$. Let us also denote by $u_i \in \{1, \ldots, m\}$ the index of the context received in round $i$. We also need to maintain a collection of $m$ sets $S_1, \ldots, S_m$, (initialized to the empty set) corresponding to $f_{\phi_1}, \ldots, f_{\phi_m}$. Our goal is to select a subset $S_j \subseteq \Omega$ for each function $f_{\phi_j}$ that maximizes its corresponding utility $f_{\phi_j}(S_j)$. Note that if $f_{\phi_j}$ were known in advance, we could simply use the greedy algorithm. However, in the interactive submodular bandit setting, we need to build up the sets $S_j$ sequentially and through interactions, as the marginal payoff of an element is only revealed after it is selected. Let $o_i \triangleq \sum_{j \leq i} 1\{u_j = u_i\}$ denote the number of occurrences of function $f_{\phi_{u_i}}$ in the first $i$ rounds. In each round, say the $i$-th with the corresponding function $f_{\phi_{u_i}}$, we need to select a new item $x_{o_i, u_i}$ from the set of items $\Omega$ and add it to $S_{u_i}$. Clearly, after including $x_{o_i, u_i}$, the set $S_{u_i}$ will be of cardinality $o_i$. For the ease of presentation, we denote $S_{u_i} \cup \{x_{o_i, u_i}\}$ by $S_{o_i, u_i}$, initialized to the empty set in the beginning, i.e., $S_{0, u_i} = \varnothing$. After selecting the item $x_{o_i, u_i}$ and given the previously selected items $S_{u_i}$, we receive $y_i$, a noisy (but unbiased) estimate of $x_{o_i, u_i}$'s marginal payoff, i.e.,

$$y_i = \Delta(x_{o_i, u_i} | S_{o_i - 1, u_i}, \phi_{u_i}) + \epsilon_i,$$

where the marginal gain $\Delta(\cdot|\cdot, \cdot) : \Omega \times 2^\Omega \times \Phi \rightarrow \mathbb{R}$ is defined as

$$\Delta(x|S, \phi) = f_\phi(S \cup \{x\}) - f_\phi(S). \tag{1}$$

We also assume that $\epsilon_i$'s are uniformly bounded noise variables that form a martingale difference sequence, i.e.,

$$\mathbb{E}[\epsilon_i | \epsilon_1, \epsilon_2, \ldots, \epsilon_{i-1}] = 0 \text{ and } |\epsilon_i| \leq \sigma \text{ for all } i \in \{1, \ldots, T\}.$$

We call $\sigma$ the *noise level*. Note that $y_i$ is the only feedback that we obtain in round $i$. It crucially depends on the *previously* selected items and *contextual information* $S_{o_i - 1, u_i}, \phi_{u_i}$. Therefore, the only avenue through which we can learn about the payoff functions $f_\phi$ (that we try to optimize over) is via noisy feedbacks $y_i$. We need to design an algorithm that minimizes the accumulated regret over the total number of $T$ rounds. Formally, we compare the performance of any algorithm in this interactive submodular bandit setting with that of the greedy algorithm with the full knowledge of the payoff functions $f_{\phi_1}, \ldots, f_{\phi_m}$.

**Algorithm 1** SM-UCB
___
**Input:** set of items $\Omega$, mean $\mu_0 = 0$, variance $\sigma_0$.
 1: Initialize $S_i \leftarrow \varnothing$ for all $i \in [m]$
 2: **for** $i = 1, 2, 3, \ldots$ **do**
 3:     select an item $x_{o_i, u_i} \leftarrow \operatorname{argmax}_{x \in \Omega} \mu_{i-1}(x) + \sqrt{\beta_i} \sigma_{i-1}(x)$
 4:     update the set $S_{u_i} \leftarrow S_{u_i} \cup \{x_{o_i, u_i}\}$
 5:     obtain the feedback $y_i = \Delta(x_{o_i, u_i} | S_{o_i - 1, u_i}, \phi_{u_i}) + \epsilon_i$
 6:     let $\mathbf{k}_i(x)$ be a vector-valued function that outputs an $i$-dimensional column vector with $j$-th entry $k((x_{o_j, u_j}, S_{o_j - 1, u_j}, \phi_{u_j}), (x, S_{o_i - 1, u_i}, \phi_{u_i}))$
 7:     let $\mathbf{K}_i$ be an $i \times i$ matrix with $(j, j')$-entry $k((x_{o_j, u_j}, S_{o_j - 1, u_j}, \phi_{u_j}), (x_{o_{j'}, u_{j'}}, S_{o_{j'} - 1, u_{j'}}, \phi_{u_{j'}}))$
 8:     update $\mathbf{y}_i \leftarrow [y_1, y_2, \ldots, y_i]^T$
 9:     let $k_i(x, x')$ be a kernel function defined as $k(x, x') - \mathbf{k}_i(x)^T (\mathbf{K}_i + \sigma^2 \mathbf{I})^{-1} \mathbf{k}_i(x')$
10:     estimate $\mu_i(x) \leftarrow \mathbf{k}_i(x)^T (\mathbf{K}_i + \sigma^2 \mathbf{I})^{-1} \mathbf{y}_i$
11:     estimate $\sigma_i(x) \leftarrow \sqrt{k_i(x, x)}$
12: **end for**
___

Suppose that by the end of the $T$-th round, an algorithm has selected $T_j$ items for the payoff function $f_{\phi_j}$; therefore the cardinality of $S_j$ by the end of the $T$-th round is $T_j$. Thus, we have $T_j = \sum_{t \leq T} 1\{u_t = j\}$ and $T = \sum_{j=1}^m T_j$. We use $S_j^*$ to denote the set that maximizes the payoff of function $f_{\phi_j}$ with at most $T_j$ elements, i.e., $S_j^* = \operatorname{argmax}_{|S| \leq T_j} f_{\phi_j}(S)$. We know that the greedy algorithm is guaranteed to achieve $(1 - 1/e) \sum_{j=1}^m f_{\phi_j}(S_j^*)$ [41] and there is no polynomial time algorithm that achieves a better approximation guarantee in general [17]. Therefore, we define the total regret of an algorithm up to round $T$ as follows:

$$\mathcal{R}_T \triangleq (1 - 1/e) \sum_{j=1}^m f_{\phi_j}(S_j^*) - \sum_{j=1}^m f_{\phi_j}(S_{T_j, j}), \tag{2}$$

which is the gap between the greedy algorithm's guarantee and the total utility obtained by the algorithm. Without any smoothness assumption over the payoff functions, it may not be possible to guarantee a sublinear regret [31]. In this paper, we make a natural assumption about the complexity of payoff functions. More specifically, we assume that the marginal payoffs, defined in (1), have a low RKHS-norm according to a kernel $k : (\Omega \times 2^\Omega \times \Phi) \times (\Omega \times 2^\Omega \times \Phi) \to \mathbb{R}$, i.e., $\|\Delta(\cdot|\cdot, \cdot)\|_k \leq B$. Note that such a kernel encodes how close two marginal payoffs are if a) the contexts $\phi_i$ and $\phi_j$ or b) the selected elements $S_i$ and $S_j$ are similar. For instance, a recommender system can leverage this information to propose an item to a user if it has observed that a user with similar features liked that item.

## 3   Main Results

In Algorithm 1 we propose SM-UCB, an interactive submodular bandit algorithm. Recall that the marginal gain function $\Delta$ has a low RKHS norm w.r.t. some kernel $k$. In each round, say the $i$-th, SM-UCB maintains the posterior mean $\mu_{i-1}(\cdot)$ and standard deviation $\sigma_{i-1}(\cdot)$ conditioned on the historical observations or context $\{(x_{o_j, u_j}, S_{o_j - 1, u_j}, \phi_{u_j}) : 1 \leq j \leq i\}$. Based on these posterior estimates, SM-UCB then selects an item $x$ that attains the highest upper confidence bound $\mu_{i-1}(x) + \sqrt{\beta_i} \sigma_{i-1}(x)$. It then receives the noisy feedback $y_i = \Delta(x_{o_i, u_i} | S_{o_i - 1, u_i}, \phi_{u_i}) + \epsilon_i$. Since $\epsilon_i$'s are uniformly bounded and form a martingale difference sequence, SM-UCB can predict the mean $\mu_i$ and standard deviation $\sigma_i$ via posterior inference in order to determine the item to be selected in the next round.

In order to bound the regret of an algorithm, we need to quantify how much information that algorithm can acquire through interactions. Let $\mathbf{y}_A$ denote a subset of noisy observations indexed by the set $A$, i.e., $\mathbf{y}_A = \{y_i | i \in A\}$. Note that any subset $A$ of noisy observations $\mathbf{y}_A$ reduces our uncertainty about the marginal gain function $\Delta$. In an extreme case, if we had perfect information (or no uncertainty) about $\Delta$, we could have achieved zero regret. We can precisely quantify this notion through what is called the *information gain* $I(\mathbf{y}_A; \Delta) = H(\mathbf{y}_A) - H(\mathbf{y}_A | \Delta)$, where $H$ denotes the Shannon entropy. In fact, by lifting the results from [31, 44] to a much more general setting, we relate the regret to the *maximum information gain* $\gamma_T$ [44] obtained after $T$ rounds and defined as

$\gamma_T \triangleq \max_{A \subseteq \Omega : |A|=T} I(\mathbf{y}_A; \Delta)$. Another important quantity that shows up in the regret bound is the *confidence parameter* $\beta_T$ (see line 3 of Algorithm 1) that needs to be chosen carefully so that our theoretical guarantee holds with high probability. In fact, the following theorem shows that SM-UCB attains a $O(\sqrt{T \beta_T \gamma_T})$ regret bound with high probability.

**Theorem 1.** *Suppose that the true marginal gain function $\Delta(\cdot|\cdot, \cdot)$ has a small RKHS norm according to some kernel $k$, i.e., $\|\Delta(\cdot|\cdot, \cdot)\|_k \leq B$. The noise variables $\epsilon_t$ satisfy $\mathbb{E}[\epsilon_t|\epsilon_1, \epsilon_2, \ldots, \epsilon_{t-1}] = 0$ for all $t \in \mathbb{N}$ and are uniformly bounded by $\sigma$. Let $\delta \in (0, 1)$, $\beta_t = 2B^2 + 300\gamma_t \ln^3(t/\delta)$ and $C_1 = 8/\log(1 + \sigma^{-2})$. Then, the accumulated regret of* SM-UCB *over $T$ rounds is as follows:*

$$\Pr\left\{\mathcal{R}_T \leq \sqrt{C_1 T \beta_T \gamma_T} + 2, \forall T \geq 1\right\} \geq 1 - \delta.$$

The proof of the above theorem is provided in the Supplementary Material. It relies on two powerful ideas: greedy selection for constrained submodular maximization [41] and upper confidence bounds of contextual Gaussian bandit [31]. If the marginal payoffs were completely known, then the greedy policy would provide a competitive solution to the optimum. However, one cannot run the greedy policy without knowing the marginal gains. In fact, there are strong negative results regarding the approximation guarantee of any polynomial time algorithm if the marginal gains are arbitrarily noisy [23]. Instead, SM-UCB relies on optimistic estimates of the marginal gains and select greedily an item with the highest upper confidence bound. By assuming that marginal gains are smooth and relying on Theorem 1 in [31], we can control the accumulated error of a greedy-like solution that relies on confidence bounds and obtain low regret. Our setting and theoretical result generalize a number of prior work mentioned below.

**Linear submodular bandit** [50]. In this setting, the objective function has the form $f(S) = \sum_{i=1}^d w_i f_i(S)$, where $f_i$'s are known submodular functions and $w_i$'s are positive *unknown* coefficients. Therefore, the marginal gain function can be written as $\Delta(x|S) = \sum_{i=1}^d w_i \Delta_i(x|S)$, where $\Delta_i(\cdot|\cdot)$'s are known functions and $w_i$'s are unknown coefficients. Let $w = (w_1, w_2, \ldots, w_d)$ denote the weight vector. Since the only unknown part of the marginal gain function is the weight vector, the space of the marginal gain function is isomorphic to the space of weight vectors, which is in fact a $d$-dimensional Euclidean space $\mathbb{R}^d$. The RKHS norm of $\Delta$ is given by some norm in $\mathbb{R}^d$; i.e., $\|\Delta\|_k \triangleq \|w\|$. The assumption in [50] that $\|w\| \leq B$ is equivalent to assuming that $\|\Delta\|_k \leq B$. Therefore, the linear bandit setting is included in our setting where the marginal gain function $\Delta$ has a special form and its RKHS norm is given by the norm of its corresponding weight vector in the Euclidean space. Also, LSBGREEDY proposed in [50], is a special case of SM-UCB (except that the feedback is delayed).

**Adaptive valuable item discovery** [47]. In this setting, the objective function has the form $f(S) = (1 - \lambda) \sum_{x \in S} g(x) + \lambda D(S)$, where $D$ is a known submodular function that quantifies the diversity of the items in $S$, $g$ is an unknown function that denotes the utility $g(x)$ for any item $x$, and $\lambda$ is a known tradeoff parameter balancing the importance of the accumulative utility and joint diversity of the items. Note that the unknown function $M(S) = \sum_{x \in S} g(x)$ is a *modular* function. Therefore, the marginal gain function has the form $\Delta(x|S) = (1 - \lambda)g(x) + \lambda D(x|S)$, where $D(x|S) \triangleq D(\{x\} \cup S) - D(S)$. The only uncertainty of $\Delta$ arises from the uncertainty about the modular function $M$. In particular, [47] assumes that the RKHS norm of $g$ is bounded. Again, our setting encompasses adaptive valuable item discovery as we consider any monotone submodular function. Moreover, GPSELECT proposed in [47], is a special case of SM-UCB.

**Contextual Gaussian bandit** [31]. This is the closest setting to ours where in each round $i$ we receive a context $\phi_i$ from the set of contexts $\Phi$ and have to choose an item $x$ from the set of items $\Omega$. We then receive a payoff $f_{\phi_i}(x) + \epsilon_t$. Note that instead of building up a set (our problem), in the contextual bandit process we simply choose a single element for each function $f_{\phi_i}$ as the main assumption is that we encounter each function only once. To obtain regret bounds it is assumed in [31] that $f$ has low norm in the RKHS associated with some kernel $k$. Again, CGP-UCB proposed in [31], is a special case of SM-UCB.

# 4 Experiments

In this section, we compare empirically the performance of SM-UCB with the following baselines:

- RANDOM. In each round, an item is randomly selected for the current payoff function $f_\phi$.
- GREEDY. It has the full knowledge of the submodular functions $f_\phi$. In each round, say the $i$-th with the corresponding function $f_{\phi_{u_i}}$, GREEDY selects the item that maximizes the marginal gain, i.e., $\operatorname{argmax}_{x \in \Omega} \Delta(x|S_{o_i-1}, \phi_{u_i})$.
- HISTORY-FREE. We run SM-UCB without considering the previously selected items. HISTORY-FREE is basically the contextual Gaussian bandit algorithm proposed in [31] whose context is the user feature.
- FEATURE-FREE. We run SM-UCB without considering the context $\phi$ of an arriving function $f$.
- CONTEXT-FREE. We run SM-UCB without considering the context or the previously selected elements. In fact, CONTEXT-FREE is basically the GP-SELECT algorithm proposed in [47].

In all of our experiments, the $m$ distinct functions $\{f_{\phi_i} : 1 \le i \le m\}$ that the algorithm encounters represent the valuation functions of $m$ users, where the context $\phi_i \in \mathbb{R}^d$ encodes users' features. Moreover, $S_i$ is the set of items that an algorithm selects for user $i \in [m]$. Instead of computing the regret, we quantify the performance of the algorithms by computing the accumulated reward $\sum_{i=1}^m f_{\phi_i}(S_i)$. Recall that the regret is given by $(1 - 1/e) \cdot OPT - \sum_{i=1}^m f_{\phi_i}(S_i)$, where $OPT = \sum_{i=1}^m f_{\phi_i}(S_i^*)$ is a constant generally hard to compute.

**Movie Recommendation** In this set of experiments, we use the MovieLens dataset[1] where a user-rating matrix $M$ is provided. The rows of $M$ represent users and the columns represent movies. The matrix $M$ contains 943 users and 1682 movies. As a preprocessing step, we apply the singular-value decomposition (SVD) to impute the missing values; the six largest singular values are kept.

In the first part of the study, we use the submatrix of $M$ that consists of $80\%$ of the users and all of the movies for training the feature vectors of movies via SVD. Let $M'$ be the submatrix of $M$ that consists of the remaining $n_{\text{user}}$ users and all of the movies; this matrix is for testing. Let $\Omega$ denote the set of movies. We consider selecting a subset of $\Omega$ to maximize the *facility-location-type* objective [30] $f(S) = \sum_{i=1}^{n_{\text{user}}} \max_{j \in S} M'_{ij}$. This objective function corresponds to a scenario without any context $\phi$ as there is only one payoff function $f$ that we are trying to maximize. Thus, FEATURE-FREE does not apply here. We use the cosine kernel $k_{\text{movie}} : \Omega \times \Omega \to \mathbb{R}$ for pairs of movies and *Jaccard* kernel $k_{\text{subset}}(S, T) = |S \cap T|/|S \cup T|$ [18] for pairs of subsets of movies, say $S$ and $T$. The composite kernel $k : (\Omega \times 2^\Omega) \times (\Omega \times 2^\Omega) \to \mathbb{R}$ is defined as $\kappa_1 k_{\text{movie}} \oplus \kappa_2 k_{\text{subset}}$, i.e., $k((u, S), (v, T)) = \kappa_1 k_{\text{movie}}(u, v) + \kappa_2 k_{\text{subset}}(S, T)$, where $\kappa_1, \kappa_2 > 0$. The results are shown in Fig. 1(a). The horizontal axis denotes the cardinality of $S$. The vertical axis denotes the function value of $f$ on the set $S$. We observe that SM-UCB outperforms all of the baselines except the practically infeasible GREEDY.

In the second part, we consider a setting where a separate subset of movies is selected for each user. We cluster the users in the dataset into 40 groups via the $k$-means algorithm and the users of the same group are viewed as identical users. The feature vector of a group of users is the mean of the feature vectors of all member users and the rating of a group is the sum of the ratings of all member users. The users are labeled as $1, 2, 3, \ldots, n'_{\text{user}}$, where $n'_{\text{user}} = 40$. Similar to the first part, the feature vectors of the users and movies are obtained via SVD. We maintain a set $S_i$ for user $i$. The objective function is $f_{\phi_i}(S) = \max_{j \in S} M''_{ij}$, where $M''_{ij}$ is user $i$'s rating for movie $j$. In addition, we also need a collective objective function that quantifies the overall performance of an algorithm for all users. It is defined as $f(S_1, S_2, \ldots, S_{n'_{\text{user}}}) = \sum_{i=1}^{n'_{\text{user}}} f_{\phi_i}(S_i)$. We assume a random arrival of users. We use the linear kernel $k_{\text{user}} : \Phi \times \Phi \to \mathbb{R}$ for pairs of users. The composite kernel $k : (\Omega \times 2^\Omega \times \Phi) \times (\Omega \times 2^\Omega \times \Phi) \to \mathbb{R}$ is defined as $\kappa_1 k_{\text{movie}} \oplus \kappa_2 k_{\text{subset}} \oplus \kappa_3 k_{\text{user}}$. In Fig. 1(b), we plot the performance of SM-UCB against other baselines. The horizontal axis denotes the number of user arrivals while the vertical axis denotes the value of the collective objective function. We

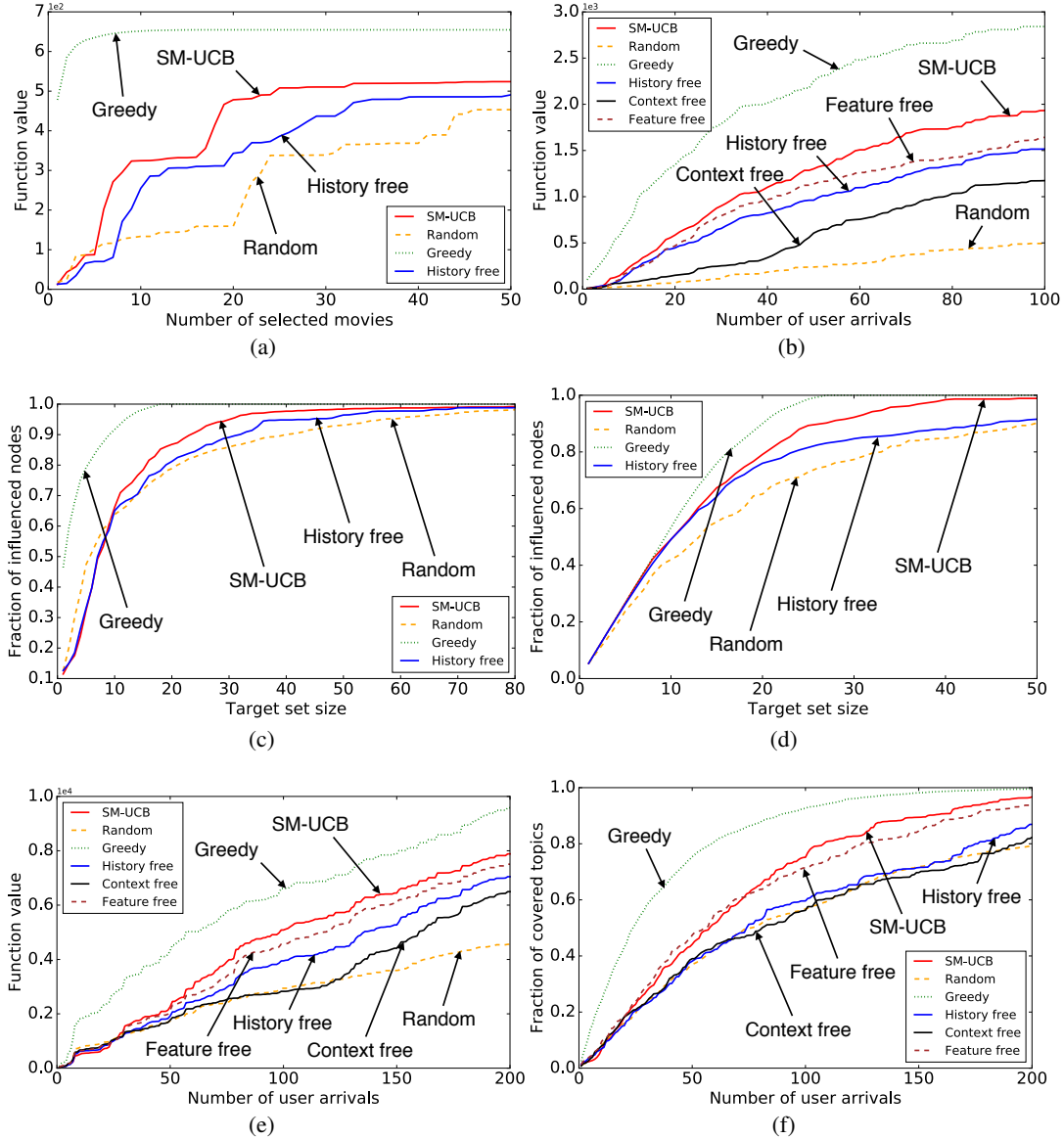

Figure 1: Figs. 1(a) and 1(b) show the results of the experiments on the MovieLens dataset. Fig. 1(a) shows how the total objective function for all users evolves as the number of selected movies increases; the algorithm recommends the same subset of movies to all users. In Fig. 1(b), we consider the situation where users arrive in a random order and we have to recommend a separate subset of movies to each user. Figs. 1(c) and 1(d) show the dependency of fraction of influenced nodes on the target set size in the Facebook network and the student network from the User Knowledge Modelling Dataset [27]. Fig. 1(e) shows how the payoff function varies as more users arrive in the Yahoo news recommender. In Fig. 1(f), we consider the personalized data summarization from Reuters corpus for arriving users. It shows the fraction of covered topics versus the number of user arrivals.

observe that SM-UCB outperforms all other baselines except GREEDY. In addition, CONTEXT-FREE that uses the least amount of information achieves a lower function value than HISTORY-FREE and FEATURE-FREE, which either leverages the information about users' features or previously selected items.

**Interactive Influence Maximization**    For this experiment, we use the Facebook network provided in [35]. The goal is to choose a subset of subjects in the network, which we call the target set, in order to maximize the number of influenced subjects. We assume that each member in the target set can influence all of her neighbors. Under this assumption, the submodular objective function is $f(S) = \left| \bigcup_{u \in S} (\mathcal{N}(u) \cup \{u\}) \right|$, where $\mathcal{N}(u)$ is the set of all neighbors of subject $u$. All the baselines, except GREEDY, have no knowledge of the underlying Facebook network or the objective function. They are only given the feature vector of each subject obtained via the NODE2VEC algorithm [21]. The kernel function $k_{\text{subject}}$ between two subjects is a linear kernel while the kernel function between subsets of subjects is the Jaccard kernel. The results are shown in Fig. 1(c). Again, SM-UCB reaches the largest influence w.r.t other baselines except for GREEDY. We ran the same idea over the 6-nearest neighbor network of randomly sampled 150 students from User Knowledge Modelling Dataset [27]. As Fig. 1(d) indicates, a similar pattern emerges.

**News Recommendation**    For this experiment, we use the Yahoo! Webscope dataset R6A[2]. The dataset provides a list of records, each containing a time stamp, a user ID, a news article ID and a Boolean value that indicates whether the user clicked on the news article that was presented to her. The feature vectors of the users and the articles are also provided. We use $k$-means clustering to cluster users into 175 groups and identify users of the same group as identical users. We form a matrix $M$ whose $(i, j)$-entry is the total number of times that user $i$ clicked on article $j$. This matrix quantifies each user's preferences regarding news articles. The objective function for user $i$ is defined as $f_{\phi_i}(S_i) = \max_{j \in S_i} M_{ij}$. The collective objective function $f$ is defined as the sum of the objective functions of all users. From the time stamps, we can infer the order in which the users arrive. We use the Laplacian kernels $k_{\text{news}} : \Omega \times \Omega \to \mathbb{R}$ and $k_{\text{user}} : \Phi \times \Phi \to \mathbb{R}$ for pairs of pieces of news and pairs of users, respectively. For a pair of subsets of news $S$ and $T$, the kernel function between them is again the Jaccard kernel. The composite kernel $k : (\Omega \times 2^{\Omega} \times \Phi) \times (\Omega \times 2^{\Omega} \times \Phi) \to \mathbb{R}$ is defined as $\kappa_1 k_{\text{news}} \oplus \kappa_2 k_{\text{subset}} \oplus \kappa_3 k_{\text{user}}$. The results are illustrated in Fig. 1(e). The horizontal axis is the number of arriving users while the vertical axis is the value of the collective objective function. Again, we observe that SM-UCB outperforms all other methods except GREEDY.

**Personalized Data Summarization**    For this experiment, we apply latent Dirichlet allocation (LDA) to the Reuters Corpus. The number of topics is set to $n_{\text{topic}} = 10$. LDA returns a topic distribution $P(i|a)$ for each article $a$ and topic $i$. Suppose that $A$ is a subset of articles. Probabilistic coverage function quantifies the degree to which a set of articles $A$ covers a topic $i$ [16], and is given by $F_i(A) = 1 - \prod_{a \in A}(1 - P(i|a))$. Each user $j$ is characterized by her $n_{\text{topic}}$-dimensional preference vector $w_j = (w_{j,1}, w_{j,2}, w_{j,3}, \ldots, w_{j,n_{\text{topic}}})$; we assume that the preference vector is $L^1$-normalized, i.e., its entries sum to 1. The personalized probabilistic coverage function for user $j$ is defined as $f_j(A) = \sum_{i=1}^{n_{\text{topic}}} w_{j,i} F_i(A)$ [16, 50]. Note that since the preference vector is $L^1$-normalized and $F_i(A) \leq 1$, we have $f_j(A) \leq 1$ for any $j$. The total average coverage function is $f(A) = \frac{1}{n_{\text{user}}} \sum_{j=1}^{n_{\text{user}}} f_j(A)$, where $n_{\text{user}} = 10$ is the number of users. Random order of user arrivals is simulated. We use the linear kernel for pairs of users and pairs of articles and use the Jaccard kernel between subsets of articles. The results are shown in Fig. 1(f). The horizontal axis is the number of user arrivals while the vertical axis is the total average coverage function $f(A)$, which characterizes the average fraction of covered topics. We observe that SM-UCB outperforms all the baselines other than GREEDY.

**Discussion**    Recall that the RKHS is a complete subspace of the $L^2$ space of functions defined on the product of the item set, its power set, and the context set. It has an inner product $(\cdot, \cdot)_k$ obeying the reproducing property: $(f, k(x, \cdot))_k = f(x)$ for all $f$ in RKHS. Functions implied by a particular kernel $k$ are always of the form $f(x) = \sum_i \alpha_i k(x_i, x)$. The bounded norm implies that $\alpha_i$ vanish quickly enough. With universal kernels like Gaussian/Laplacian kernels, such functions are dense (according to sup-norm) in the space of continuous functions.

In three sets of experiments (movie recommendation, influence maximization, data summarization) we used the linear and cosine kernels for items and users, and the Jaccard kernel for subsets of items. In fact, the Jaccard kernel is a widely used metric that quantifies the similarity between subsets of selected items. Moreover, the linear and cosine kernels between items and users capture the simplest form of interactions. In contrast to the the above three experiments, in the news recommendation

application, we chose the Laplacian kernel for the following reason. The features provided in the dataset have highly heterogeneous norms. If we use the linear kernel, the inner product between a short vector and a close-by vector with a small norm will be easily dominated by the inner product with a vector with a large norm. We used the Laplacian kernel to circumvent this problem and put more weight on nearby vectors even if they have small norms.

## 5  Related Work

Originally, Auer et al. [2] proposed UCB policies for the multi-armed bandit (MAB) problem which exhibits the exploration-exploitation tradeoff and achieves an $O(\sqrt{T})$ regret. In the literature, there are many variants of the multi-armed bandit problem and corresponding solutions, for example, EXP3 algorithm for adversarial bandits [3], LINUCB for stochastic contextual bandits [36, 12], and a family of UCB-based policies for infinitely many-armed bandit [49]. Chen et al. [11] considered the combinatorial MAB problem where the unit of play is a super arm and base arms can be probabilistically triggered. For a comprehensive survey on bandit problems, we refer the interested reader to [9].

Srinivas et al. [44] studied the Gaussian process (GP) optimization problem in the bandit setting. They assumed that the objective function $f$ is either sampled from a Gaussian process or resides in a reproducing kernel Hilbert space (RKHS). Given a subset of items $S \subseteq \Omega$, the total utility is $\sum_{x \in \Omega} f(x)$. Under either the GP model or the RKHS assumption, they showed that their proposed GP-UCB algorithm achieves an $O(\sqrt{T})$ regret bound. It is noteworthy to mention that their bound also relies on the maximum information gain. Based on [44], Krause and Ong [31] further investigated the contextual Guassian process bandit optimization and their proposed algorithm CGP-UCB achieves a similar regret bound. Lin et al. [37] addressed an online learning problem where the input to the greedy algorithm is stochastic with unknown parameters and the algorithm receives semi-bandit feedbacks. Their algorithm can also be applied to submodular functions. However, there are several major differences between their work and ours: Firstly, they assume that the objective functions are drawn from a predetermined but unknown distribution, while our work applies to any set of submodular functions; secondly they assume bounded submodular functions while we have no such assumptions; thirdly, their work did not have the notion of context. They optimize the expected objective function while we optimize objective functions with different contexts simultaneously. Streeter and Golovin [45] studied the online maximization problem of submodular functions. Yue and Guestrin [50] studied the linear submodular bandit problem where they assumed that the unknown submodular function is a linear combination of multiple *known* submodular functions. The only uncertainty in their setting is the unknown positive coefficients of each known submodular function. They proposed LSBGREEDY that achieves a similar $O(\sqrt{T})$ regret bound. Beyond unconstrained sequential decision problems, Zhou et al. [52] considered online maximization of list submodular functions under a knapsack constraint.

Our key contribution in this paper is that the notion of contextual regret that we bound is much more challenging than the typical notion: Our actions are affecting the future contexts experienced, and we compete with policies that are aware of this fact and can plan for it. This is qualitatively different from any prior analysis. More specifically, we need to build up a subset of items/actions as we encounter a valuation function multiple times. This is a non-trivial task as not only the functions are unknown, the marginal gains are also noisy. Moreover, the choices we make can affect the future. Our positive results can be seen in light of very recent negative results in [23] that indicates submodular optimization is hard when function evaluations are noisy. We show that the UCB-based algorithm can be naturally combined with the greedy selection policy to provide sublinear regret. To the best of our knowledge the analysis is new.

### Acknowledgements

This research was supported by DARPA Young Faculty Award (D16AP00046), grant SCADAPT and ERC StG.

## Footnotes

[1]https://grouplens.org/datasets/movielens/

[2]`http://webscope.sandbox.yahoo.com/`

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
