[Supplementary Material]

# Appendix

### Proof of Theorem 1

We define the regret when the algorithm selects the $i$th item for user $j$

$$r_{i,j} = \sup_{x \in \Omega} \Delta(x|S_{i-1,j}, \phi_j) - \Delta(x_{i,j}|S_{i-1,j}, \phi_j).$$

At this time, the accumulated regret for user $j$ by the end of the $i$th selection is

$$R_{i,j} = \sum_{l=1}^{i} r_{l,j}$$

and the algorithm has selected

$$S_{i,j} = \{x_{1,j}, x_{2,j}, \ldots, x_{i,j}\}$$

for user $j$.

In light of the monotonicity of $f_{\phi_j}$, we have

$$\begin{aligned}
f_{\phi_j}(S_j^*) &\leq f_{\phi_j}(S_j^* \cup S_{i,j}) \\
&\leq f_{\phi_j}(S_{i,j}) + \sum_{v \in S_j^*} \Delta(v|S_{i,j}, \phi_j) \\
&\leq f_{\phi_j}(S_{i,j}) + T_j \sup_{x \in \Omega} \Delta(x|S_{i,j}, \phi_j) \\
&= f_{\phi_j}(S_{i,j}) + T_j(r_{i+1,j} + \Delta(x_{i+1,j}|S_{i,j}, \phi_j)) \\
&= f_{\phi_j}(S_{i,j}) + T_j(R_{i+1,j} - R_{i,j} + f_{\phi_j}(S_{i+1,j}) - f_{\phi_j}(S_{i,j})),
\end{aligned} \tag{3}$$

where Eq. (3), we use the definition of $r_{i+1,j}$, i.e.,

$$r_{i+1,j} = \sup_{x \in \Omega} \Delta(x|S_{i,j}, \phi_j) - \Delta(x_{i+1,j}|S_{i,j}, \phi_j),$$

which yields

$$r_{i+1,j} + \Delta(x_{i+1,j}|S_{i,j}, \phi_j) = \sup_{x \in \Omega} \Delta(x|S_{i,j}, \phi_j).$$

Therefore, we have

$$f_{\phi_j}(S_j^*) - f_{\phi_j}(S_{i,j}) \leq T_j(R_{i+1,j} - R_{i,j} + f_{\phi_j}(S_{i+1,j}) - f_{\phi_j}(S_{i,j})).$$

Let $\delta_{i,j} = f_{\phi_j}(S_j^*) - f_{\phi_j}(S_{i,j})$. Then we have

$$\delta_{i,j} - \delta_{i+1,j} = (f_{\phi_j}(S_j^*) - f_{\phi_j}(S_{i,j})) - (f_{\phi_j}(S_j^*) - f_{\phi_j}(S_{i+1,j})) = f_{\phi_j}(S_{i+1,j}) - f_{\phi_j}(S_{i,j}).$$

Therefore, we obtain that

$$\delta_{i,j} \leq T_j(R_{i+1,j} - R_{i,j} + f_{\phi_j}(S_{i+1,j}) - f_{\phi_j}(S_{i,j})) \leq T_j(R_{i+1,j} - R_{i,j} + \delta_{i,j} - \delta_{i+1,j})$$

which entails

$$\delta_{i+1,j} \leq R_{i+1,j} - R_{i,j} + (1 - 1/T_j)\delta_{i,j}.$$

Hence for all $i$, we have

$$\delta_{i,j} \leq R_{i,j} - R_{i-1,j} + (1 - 1/T_j)\delta_{i-1,j}.$$

Note that $f_{\phi_j}$ is normalized. Therefore,

$$\delta_{0,j} = f_{\phi_j}(S_j^*) - f_{\phi_j}(S_{0,j}) = f_{\phi_j}(S_j^*) - f_{\phi_j}(\varnothing) = f_{\phi_j}(S_j^*).$$

We recursively solve it with respect to $\delta_{i,j}$ and obtain

$$\delta_{i,j} \leq \sum_{l=1}^{i}(1-\frac{1}{T_j})^{l-1}(R_{i-l+1,j}-R_{i-l,j})+(1-\frac{1}{T_j})^i\delta_{0,j}$$

$$\leq \sum_{l=1}^{i}(1-\frac{1}{T_j})^{l-1}(R_{i-l+1,j}-R_{i-l,j})+(1-\frac{1}{T_j})^i f_{\phi_j}(S_j^*)$$

$$= \sum_{l=0}^{i-1}(1-\frac{1}{T_j})^{i-l-1}(R_{l+1,j}-R_{l,j})+(1-\frac{1}{T_j})^i f_{\phi_j}(S_j^*)$$

$$= \sum_{1\leq l\leq i}(1-\frac{1}{T_j})^{i-l}R_{l,j} - \sum_{0\leq l\leq i-1}(1-\frac{1}{T_j})^{i-l-1}R_{l,j}+(1-\frac{1}{T_j})^i f_{\phi_j}(S_j^*)$$

$$= R_{i,j}-(1-\frac{1}{T_j})^{i-1}R_{0,j} + \sum_{1\leq l\leq i-1}\left[(1-\frac{1}{T_j})^{i-l}-(1-\frac{1}{T_j})^{i-l-1}\right]R_{l,j}+(1-\frac{1}{T_j})^i f_{\phi_j}(S_j^*)$$

$$= R_{i,j}-\frac{1}{T_j}\sum_{1\leq l\leq i-1}(1-1/T_j)^{i-l-1}R_{l,j}+(1-\frac{1}{T_j})^i f_{\phi_j}(S_j^*)$$

$$\leq R_{i,j}-\frac{1}{T_j}\sum_{1\leq l\leq i-1}(1-\frac{1}{T_j})^{i-l-1}R_{l,j}+e^{-i/T_j}f_{\phi_j}(S_j^*).$$

Using $\delta_{i,j}=f_{\phi_j}(S_j^*)-f_{\phi_j}(S_{i,j})$, we obtain the following equation after some simple algebra

$$(1-e^{-i/T_j})f_{\phi_j}(S_j^*) \leq f_{\phi_j}(S_{i,j})+R_{i,j}-\frac{1}{T_j}\sum_{1\leq l\leq i-1}(1-1/T_j)^{i-l-1}R_{l,j} \leq f_{\phi_j}(S_{i,j})+R_{i,j}.$$

Let $i=T_j$ in the above equation and we immediately have

$$(1-1/e)f_{\phi_j}(S_j^*) \leq f_{\phi_j}(S_{T_j,j})+R_{T_j,j},$$

which entails

$$(1-1/e)f_{\phi_j}(S_j^*)-f_{\phi_j}(S_{T_j,j}) \leq R_{T_j,j}.$$

Summing the above inequality over $j$ yields

$$\mathcal{R}_T = (1-1/e)\sum_{j=1}^{m}f_{\phi_j}(S_j^*)-\sum_{j=1}^{m}f_{\phi_j}(S_{T_j,j})$$

$$\leq \sum_{j=1}^{m}R_{T_j,j}$$

$$= \sum_{j=1}^{m}\sum_{i=1}^{T_j}r_{i,j}.$$

As shown above, we bound the regret $\mathcal{R}_T$ by $\sum_{j=1}^{m}\sum_{i=1}^{T_j}r_{i,j}$. If we define the regret of each iteration (say, the $i$th iteration) as

$$r_i = \sup_{x\in\Omega}\Delta(x|S_{o_i-1,u_i},\phi_{u_i})-\Delta(x_{o_i,u_i}|S_{o_i-1,u_i},\phi_{u_i}),$$

we have

$$\sum_{i=1}^{T} r_i = \sum_{i=1}^{T} (\sup_{x \in \Omega} \Delta(x|S_{o_i-1,u_i}, \phi_{u_i}) - \Delta(x_{o_i,u_i}|S_{o_i-1,u_i}, \phi_{u_i}))$$

$$= \sum_{i=1}^{T} r_{o_i,u_i}$$

$$= \sum_{j=1}^{m} \sum_{1 \le i \le T, u_i=j} r_{o_i,u_i}$$

$$= \sum_{j=1}^{m} \sum_{1 \le i \le T, u_i=j} r_{o_i,j}$$

$$= \sum_{j=1}^{m} \sum_{i=1}^{T_j} r_{i,j}.$$

Therefore, $\mathcal{R}_T$ is bounded by $R_T$, where $R_T$ is defined as $\sum_{i=1}^{T} r_i$.

We can model the problem that we consider in this paper as a contextual bandit problem, where $(S_{o_i-1,u_i}, \phi_{u_i})$ is the context at the $i$th iteration and the element to be selected is the action. By Theorem 1 in [31] (specifically, the third assumption in the theorem statement applies here), we have

$$\Pr\left\{R_T \le \sqrt{C_1 T \beta_T \gamma_T} + 2, \forall T \ge 1\right\} \ge 1 - \delta.$$

Hence, we conclude that

$$\Pr\left\{\mathcal{R}_T \le \sqrt{C_1 T \beta_T \gamma_T} + 2, \forall T \ge 1\right\} \ge 1 - \delta$$

since $\mathcal{R}_T \le R_T$ for all $T \ge 1$.