[Reviews · NeurIPS 2017]

Reviewer 1



The papers builds on CGP-UCB. Although one can apply CGP-UCB to a submodular set function of a growing set of selected actions, the original notion of regret is about individual actions of adding elements to the sets. The paper manages to express the theoretical results for CGP-UCB for a notion of regret that is meaningful for the interactive submodular setting, comparing the final values of each set to the (1 - 1/e) approximation guarantee of greedy item selection. Similarly, the main assumption for the regret bound is expressed in terms of the marginal gains for the submodular functions in a RKHS. Experimental results are provided on a variety of problems. The paper is well written and generalizes upon multiple recent works. This appears to have many practical applications. Regret is not explicitly mentioned in the experiments section. On some of them, \sqrt{T} regret is not too obvious as the gap between greedy and SM-UCB appears constant. This may be a small-sample effect. There is also no exploration on the influence of the kernel on the performance in those datasets (e.g., how "low" is actually the norm of the marginal gain functions with respect to the reproducing kernel in those cases? Does this correlate with better performance in practice?) The paper repeatedly mentions that the proposed method outperforms all prior art. This is a bit misleading, as the only such baseline is "random subset selection". Other baselines are introduced by dumbing-down the proposed method, although this does not quite qualify as prior art. Remarks L48: "of the different between" L226: "Thr rows"

Reviewer 2



Pros: - The paper combines submodular function maximization with contextual bandit, and propose the problem of interactive submodular bandit formulation and SM-UCB solution. It includes several existing studies as special cases. - The empirical evaluation considers several application contexts and demonstrates the effectiveness of the SM-UCB algorithm. Cons: - The discussion on the key RKHS condition is not enough, and its applicability for actual applications is unclear. More importantly, there is no discussion about constant B for the RKHS norm for various applications, and also the maximum information gain \gamma_t, but both of them are needed in the algorithm. - The technical analysis is heavily dependent on prior study, in particular [33]. The additional technical contribution is small. The paper studies a contextual bandit learning setting, where in each round a context \phi is presented, which corresponds to an unknown submodular set function f_\phi. The user needs to select one item x to add to the existing selected set S_\phi, with the purpose of maximizing the marginal gain of x given S_\phi. After selecting x, the user obtains a noisy feedback of the marginal gain. The regret is comparing against the (1-1/e) fraction of the optimal offline solution, when all the submodular functions for all contexts are known. The key assumption to make the learning work is that the marginal contribution \Delta in terms of the subset S, the new item x, and the context \phi, has a bounded RKHS norm. Intuitively, this norm makes sure that similar contexts, similar subsets, and similar item would give similar marginal results. The authors propose the SM-UCB algorithm to solve the problem. The proposed algorithm and its regret analysis is heavily based on the prior work on contextual Gaussian process bandit optimization [33]. In fact, it looks like all the authors need to do is to treat existing subset S already selected as part of the context, and then plugging it into the algorithm and the analysis of [33]. The proof of Theorem 1 given in the supplementary material is only to connect their current regret definition with the regret definition of [33], using the submodularity property. Then they just apply Theorem 1 of [33] to obtain the result. Therefore, from the technical point of view, the paper's new contribution is small. [Note: After reading the authors' feedback, I get a bit clearer understanding, in that the difference in model from [33] seems to be that the current action may affect future contexts. However, I am still not quite sure how much the analysis differs from [33], since there is only a two-and-a-half page analysis in the supplementary material, and it mostly relies on [33] to achieve the main result. Please clarify the technical contribution of the current paper, and what are the novelty in the analysis that differs from [33]] Thus, the main contribution of the paper, in my view, is to connecting submodular maximization with prior work on contextual bandit and give the formulation and the result of interactive submodular bandit problem. Their empirical evaluation covers several application cases and demonstrate advantages of the AM-UCB algorithm, giving readers some concrete examples on how to apply their general results in different contexts. However, one important thing the authors did not discuss much is the key assumption on the bounded RKHS norm. First, RKHS norm should be better defined and explained. Second, its implication on several application contexts should be discussed. Third, the choice of kernel functions should be discussed. More importantly, the bound B should be evaluated for the applications they discuss and use in the experiment section. This is because this complexity measure B is not only used in the regret bound, but also appears as part of \beta_t in the algorithm. Similar situation occurs for the maximum information gain paramter \gamma_t. The authors do not discuss at all how to obtain these parameters for the algorithm. Without these parameters, it is not even clear how the authors evaluate their SM-UCB algorithm in the experiments. This is a serious ommission, and the readers would not be able to apply their algorithm, or reproduce their experiments without knowing how to properly set B and \gamma_t. Furthermore, In the empirical evaluation section, the authors use linear kernel and Jaccard kernel, but why these are reasonable ones? What are their implications to the application context? What are the other possible kernels to use? In general, I do not have a good idea how to pick a good kernel function for an application, and what are the impact and limitations of this assumption. I think this should be the key contribution of the paper, since the other technical contribution is minor, but the authors fail to provide satisfactory answers in this regard. Other minor comments: - line 133, "T_i" --> "T_j" - line 164, I(y_A | \Delta) --> I(y_A; \Delta) - Related work It would be nice if the authors could compare their approach with other related approaches combining multi-armed bandit with combinatorial optimization tasks, such as a) combinatorial multi-armed bandit with application to influence maximization and others: Combinatorial Multi-Armed Bandit and Its Extension to Probabilistically Triggered Arms Wei Chen, Yajun Wang, Yang Yuan, Qinshi Wang, in Journal of Machine Learning Research, 2016 b) Online influence maximization. Siyu Lei, Silviu Maniu, Luyi Mo, Reynold Cheng, and Pierre Senellart. In KDD, 2015. c) Stochastic Online Greedy Learning with Semi-bandit Feedbacks. Tian Lin, Jian Li, Wei Chen, In NIPS'2015

Reviewer 3



This paper proposes a general framework for interactive submodular optimization with bandit feedback. Specifically, the algorithm must maximize over a collection of submodular functions. At each iteration, the algorithm receives context corresponding to which submodular function it is currently maximizes. The algorithm selects an item to add the set for that function, and receives a noise corrupted feedback corresponding to the marginal gain. Furthermore, one assumes a kernel that captures the correlations between (context, set, action) pairs, and the regret bounds depend on maximum information gain of this kernel. One uncertainty I have about this submission is its overall novelty. What do the authors view as the novel part of their contribution? Is it the unified framework? Is it the analysis? From my vantage point, the unified framework, while nice, is not that surprising. The proof has some surprising elements to it, namely that one can allow the future feedback to be influenced by the current chosen actions. That was not entirely obvious (to me) before. A minor point -- I don't think the proposed approach fully generalizes the Linear Submodular Bandits setting. In that work, the algorithm must select L items before receiving any feedback, and the proposed framework doesn't deal with such delayed feedback. However, I think the generalization is straightforward. The above minor point also accentuates my uncertainty about the novelty of the contribution. Each way that the proposed framework generalizes previous work is, in my opinion, relatively incremental. One can keep adding more pieces to the framework and claim a broader generalization, but it's not clear that this paper shifts the way people think about research in this direction. I'm happy for the authors to correct my misconception in this regard, if indeed I am being shortsighted here. The notation is kind of sloppy. For example, u_i is not formally defined. The actual iterative procedure is also not well specified. For example, what is the step-by-step sequence of interactions between the algorithm and the environment? Does it first receive a u_i, then a phi_i, then chooses an item x? One can sort of infer this from the algorithm pseudo-code, but it would've been much clearer to formally define the problem interaction setup. ** RESPONSE TO AUTHOR FEEDBACK ** I thank the authors for their response. For revising the paper, I think the comparison w.r.t. [33] should be made more explicit in terms of the exact technical contributions